# Translational buffering by ribosome stalling in upstream open reading frames

**Ty A. Bottorff[1,2], Heungwon Park[1], Adam P. Geballe[3]\*, Arvind Rasi Subramaniam[1,2]\***

**1** Basic Sciences Division and Computational Biology Program of the Public Health Sciences Division, Fred Hutchinson Cancer Center, Seattle, Washington, United States of America, **2** Biological Physics, Structure and Design Graduate Program, University of Washington, Seattle, Washington, United States of America, **3** Human Biology and Clinical Research Divisions, Fred Hutchinson Cancer Center, Seattle, Washington, United States of America

\* ageballe@fredhutch.org (APG); rasi@fredhutch.org (ARS)

**Data Availability Statement:** All data and code associated with the manuscript is publicly available at https://github.com/rasilab/bottorff_2022.

**Funding:** This work was supported by grants from the National Institute of General Medical Sciences

## Abstract

Upstream open reading frames (uORFs) are present in over half of all human mRNAs. uORFs can potently regulate the translation of downstream open reading frames through several mechanisms: siphoning away scanning ribosomes, regulating re-initiation, and allowing interactions between scanning and elongating ribosomes. However, the consequences of these different mechanisms for the regulation of protein expression remain incompletely understood. Here, we performed systematic measurements on the uORF-containing 5′ UTR of the cytomegaloviral *UL4* mRNA to test alternative models of uORF-mediated regulation in human cells. We find that a terminal diproline-dependent elongating ribosome stall in the *UL4* uORF prevents decreases in main ORF protein expression when ribosome loading onto the mRNA is reduced. This uORF-mediated buffering is insensitive to the location of the ribosome stall along the uORF. Computational kinetic modeling based on our measurements suggests that scanning ribosomes dissociate rather than queue when they collide with stalled elongating ribosomes within the *UL4* uORF. We identify several human uORFs that repress main ORF protein expression via a similar terminal diproline motif. We propose that ribosome stalls in uORFs provide a general mechanism for buffering against reductions in main ORF translation during stress and developmental transitions.

## Author summary

Life requires proteins for nearly all functions. mRNA molecules relay information from the DNA code to protein-making molecular machines called ribosomes. Ribosomes load onto mRNA molecules and translate sections of the code, termed open reading frames, to make proteins. Cells' needs for proteins change depending on the cell type and growth environment. Thus, protein synthesis is a highly regulated process. One way for cells to regulate protein synthesis is to vary the rate at which ribosomes load onto mRNA molecules. Unexpectedly, we found that increasing the rate of ribosome loading onto mRNA molecules can decrease protein synthesis. This unexpected result can arise in mRNA molecules that have multiple open reading frames from which multiple distinct proteins are

(GM119835, to ARS), the National Science Foundation (1846521, to ARS) and the National Institute of Allergy and Infectious Diseases (AI156152, to APG), and a National Science Foundation graduate fellowship (2140004, to TAB). This research was supported by the Genomics Shared Resource of the Fred Hutch/University of Washington Cancer Consortium (P30 CA015704) and Fred Hutch Scientific Computing (NIH grants S10-OD-020069 and S10-OD-028685). The funders had no role in study design, data collection and analysis, decision to publish, or preparation of the manuscript.

**Competing interests:** The authors have declared that no competing interests exist.

made. If there are sequences that stall ribosomes within the first encountered open reading frame on an mRNA molecule, then we find that increasing ribosome loading can decrease protein synthesis at the second open reading frame. Our findings can be explained if trailing ribosomes that collide with a stalled ribosome within the first encountered open reading frame dissociate from, rather than queue on, the mRNA molecule. Our findings have implications for stress-responsive mRNA molecules whose second open reading frames are preferentially translated during cellular stress when the ribosome loading rate is reduced.

## Introduction

About half of human mRNAs have at least one upstream open reading frame (uORF) in their 5′ untranslated region [1–3]. Ribosome profiling studies estimate that at least twenty percent of these uORFs are actively translated [4,5]. uORFs can regulate gene expression via the biological activity of the uORF peptide, but they also often *cis*-regulate translation of the downstream main ORF [6,7]. Despite having poor initiation sequence contexts, many eukaryotic uORFs repress main ORF translation [1,3,4,7–11]. uORF mutations are implicated in several human diseases via changes to main ORF translation [12,13]. For example, uORF mutations in oncogenes and tumor suppressors can act as driver mutations in cancer [14,15].

uORFs can regulate translation via a variety of molecular mechanisms. uORFs can constitutively repress translation by siphoning away scanning ribosomes from initiating at downstream main ORFs. Multiple uORFs can interact together to regulate the re-initiation frequency at the main ORF. For example, uORFs in the *S. cerevisiae GCN4* mRNA and the homologous human *ATF4* mRNA render main ORF translation sensitive to cellular levels of the eIF2α-GTP-tRNA$_{Met}$ ternary complex [16,17]. Although the initiation rate usually limits translation [18–20], inefficient elongation or termination on uORFs can also regulate protein expression by preventing scanning ribosomes from reaching the main ORF [21–26]. Inefficient elongation can be driven by the nascent uORF peptide [27,28], poorly translated codons in the uORF [29,30], or small molecules such as amino acids or polyamines [23,24]. Further, interactions between scanning and elongating ribosomes on uORFs may cause dissociation of scanning ribosomes or enhanced initiation at start codons [23,31,32].

Despite the plethora of proposed uORF regulatory mechanisms, their implications for the regulation of protein expression are not clear. For example, are some uORF regulatory mechanisms more effective than others at repressing protein expression across a wide range of biochemical parameters? How do uORFs alter the response of main ORF translation to changes in cellular and environmental conditions? Answering these questions requires a joint accounting of how the different steps of translation, such as initiation, scanning, and elongation, together influence the overall rates of uORF and main ORF translation. Since it is not straightforward to monitor the rates of individual steps of translation [33], indirect measurements of protein expression are often necessary to infer the underlying mechanism of uORF-mediated regulation. Such inference requires rigorous kinetic models of uORF regulation that make testable experimental predictions for the effects of genetic mutations on protein expression.

Computational kinetic modeling has been widely used to study mechanisms of translational control [34]. Quantitative modeling of uORF translation has been used to support the regulated re-initiation model for the *GCN4* mRNA [35–37]. A computational model predicted that elongating ribosomes can dislodge leading scanning ribosomes on uORFs and confer stress resistance to protein expression [38]. However, these models have not been compared against

alternative models of uORF regulation that predict queuing or dissociation of scanning ribosomes upon collision with paused elongating ribosomes [21,23]. A critical barrier to such comparison has been the lack of a computational framework for the specification and simulation of different kinetic models of uORF-mediated translational regulation. Such a computational framework is necessary for the identification of unique experimental signatures of each proposed model and for their comparison with experimental measurements. Even though simulation code has been made available in many computational studies of mRNA translation [18,38], it is often highly tailored for specific models and cannot be easily modified to consider alternative regulatory mechanisms.

Here, we use experimental measurements on the well-studied uORF-containing 5′ UTR of the human cytomegaloviral *UL4* mRNA to test different kinetic models of uORF-mediated translational control [21]. The second uORF (henceforth uORF2) in the *UL4* 5′ UTR contains a terminal diproline motif that stalls 80S ribosomes by disrupting peptidyl transferase center activity [27,28]. For systematic model comparisons, we rely on a recent computational framework that allows easy specification and efficient simulation of arbitrary kinetic models of translational control [39]. Using this experimentally-integrated modeling approach, we find that the presence of 80S stalls in uORF2 of *UL4* 5′ UTR confers resistance (henceforth called buffering) of main ORF translation to reduced ribosome loading on the mRNA. Modeling suggests that collisions of scanning ribosomes with the stalled 80S ribosome confer this buffering behavior. Experimental variation of the distance between the uORF2 start codon and the elongating ribosome stall supports a kinetic model in which scanning ribosomes dissociate rather than queue upon colliding with the 80S stall. We also identify several human uORFs that have repressive terminal diproline motifs similar to the *UL4* uORF2 80S stall. We propose that ribosome stalls in uORFs enable buffering of main ORF protein expression against reduced ribosome loading across cellular and environmental transitions. Together, our results illustrate the value of experimentally-integrated kinetic modeling for the comparison of different uORF regulatory mechanisms and the identification of novel experimental signatures from complex molecular interactions.

## Results

### Models of uORF regulation of main ORF translation

We surveyed five previously proposed models of uORF regulation of main ORF translation (Fig 1). We tested these models using a combination of computational modeling and experimental reporter assays. In the constitutive repression model [9] (Fig 1A), uORFs siphon away scanning ribosomes from the main ORF since re-initiation is usually infrequent [40–43]. In the 80S-hit dissociation model [38] (Fig 1B), elongating ribosomes that hit downstream scanning ribosomes cause the 3′ scanning ribosomes to dissociate from the mRNA. In the queuing-mediated enhanced repression model [23] (Fig 1C), a stalled elongating ribosome within the uORF allows upstream scanning ribosomes to queue in the 5′ region. This queuing can bias scanning ribosomes to initiate translation at the uORF start codon rather than leaky scan past it. In the collision-mediated 40S dissociation model [31,32] (Fig 1D), scanning ribosomes instead dissociate if they collide with a 3′ stalled elongating ribosome.

Lastly, in the regulated re-initiation model [16,44,45] (Fig 1E), for example in the *GCN4* (*S. cerevisiae* homolog of human *ATF4*) mRNA, translation of the first uORF is followed by re-initiation at either a second downstream uORF or the main ORF depending on the stress status of the cell. After termination at the first uORF stop codon, scanning ribosomes must reacquire a new eIF2α-GTP-tRNA$_{Met}$ ternary complex (TC) before re-initiating at a downstream start codon. The time to reacquire a new TC correlates with the proportion of phosphorylated

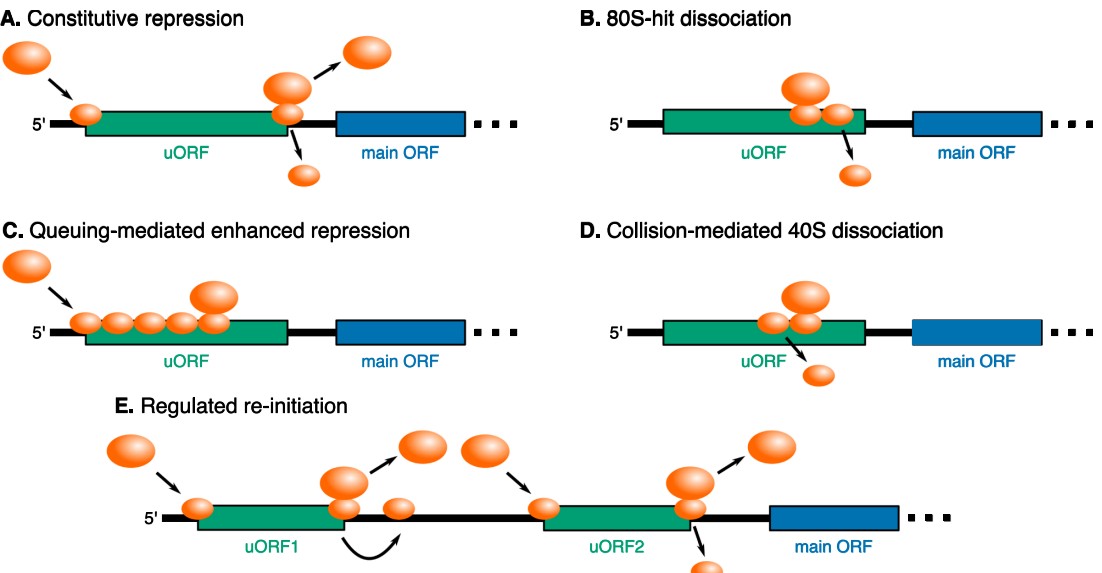

**Fig 1. Models of uORF regulation considered in this study. (A)** *Constitutive repression*. The uORF constitutively siphons away a proportion of scanning ribosomes from the main ORF. **(B)** *80S-hit dissociation*. Elongating ribosomes that collide with 3′ scanning ribosomes cause the leading scanning ribosome to dissociate from the mRNA. **(C)** *Queuing-mediated enhanced repression*. Scanning or elongating ribosomes form a queue behind a 3′ stalled elongating ribosome. If the queue correctly positions a scanning ribosome at the uORF start codon, then the proportion of scanning ribosomes that initiate translation at the uORF start codon increases. **(D)** *Collision-mediated 40S dissociation*. Scanning ribosomes that collide with a 3′ stalled ribosome dissociate from the mRNA. **(E)** *Regulated re-initiation*. Ribosomes initiate translation at the first uORF start codon, and scanning continues after termination at the stop codon of the first uORF. Ribosomes re-initiate at the main ORF start codon or the second downstream uORF start codon when phosphorylated eIF2α levels are high or low, respectively. The schematic is depicted in a low phosphorylated eIF2α state.

eIF2α. Therefore, when cells are not stressed and the proportion of phosphorylated eIF2α is lower, translation of the first uORF is followed by re-initiation at the second downstream uORF start codon. Alternatively, when cells are stressed and the proportion of phosphorylated eIF2α is higher, translation of the first uORF is instead followed by re-initiation at the main ORF start codon.

## Experimental system for testing different models of uORF-mediated translational regulation

To differentiate between proposed models of uORF regulation (Fig 1), we used the well-studied human cytomegaloviral *UL4* uORF2 [31] as an experimental model (Fig 2A). uORF2 represses main ORF translation via an elongating ribosome stall that is dependent on the uORF2 peptide sequence [21] (Fig 2A, irrelevant uORFs boxed in white, key uORF2 boxed in green). The two C-terminal proline residues, regardless of codon usage, in uORF2 are necessary for the elongating ribosome stall [32]. These residues are poor substrates for nucleophilic attack to generate a peptide bond and also reorient the ribosomal peptidyl transferase center to reduce termination activity [28]. Termination activity is further reduced through interactions between the uORF2 nascent peptide and the GGQ motif within eRF1 [46]. Even though the A-site of the uORF2-stalled ribosome is occupied by a stop codon, we refer to it as an elongating ribosome stall since they are functionally equivalent for the purposes of this study. This terminology is also inclusive of elongation stalls within other uORFs [22,23,47–51]. The 5′ leader region preceding the *UL4* coding sequence contains two other uORFs besides uORF2. uORF1

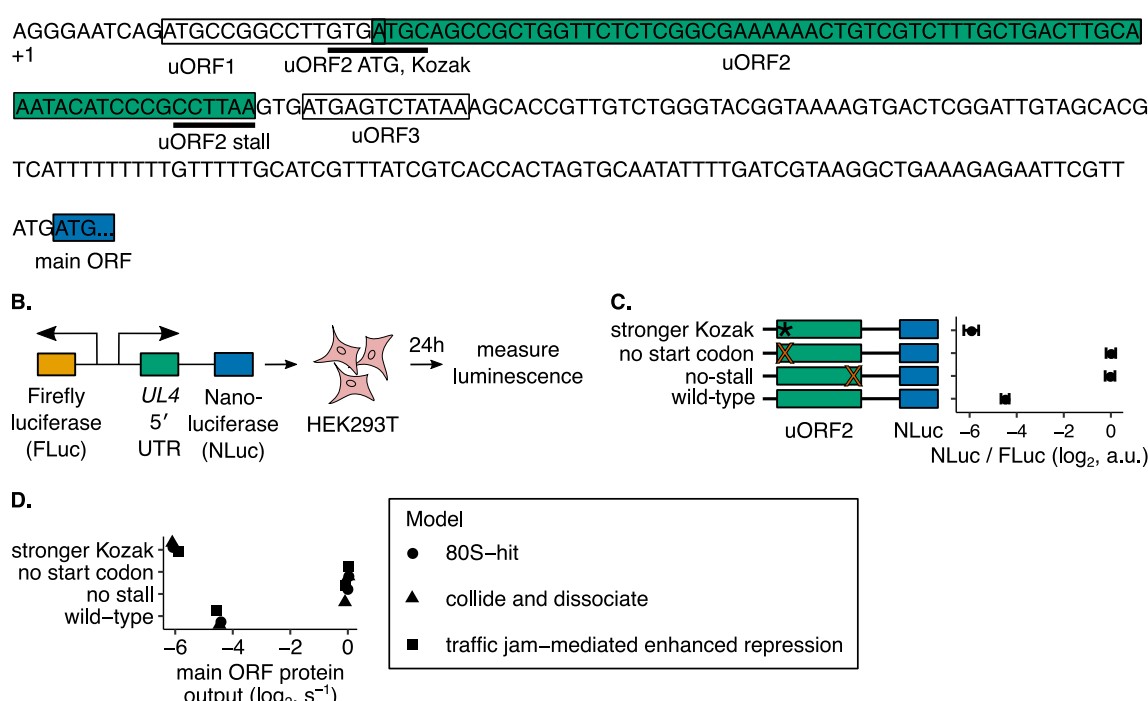

**Fig 2. An experimental and computational platform for assessing uORF-mediated regulation of main ORF translation. (A)** The 236 nt 5′ UTR of *UL4* mRNA from human cytomegalovirus contains 3 uORFs. The terminal proline and stop codons of uORF2 at which the P- and A-sites of the stalled ribosome are positioned are highlighted as uORF2 stall. **(B)** A dual-luciferase reporter system for measuring 5′ UTR repressiveness in HEK293T cells. FLuc signal serves as an internal control for transfection efficiency. **(C)** The reporter system recapitulates the known elongating ribosome stall-dependent repression of protein expression by the *UL4* uORF2 [21]. The indicated mutations improve the uORF2 Kozak context (ACCATGG instead of GTGATGC), remove the start codon (ACC instead of ATG), or remove the elongating ribosome stall by mutating the terminal proline codon to an alanine codon (GCT instead of CCT). Error bars show standard error of mean NLuc / FLuc ratios over 3 biological replicates. Data are normalized to the no-uORF start codon control. **(D)** Computationally predicted uORF regulation in the 80S-hit dissociation, queuing-mediated enhanced repression, and collision-mediated 40S dissociation models. Data are normalized to the no-uORF start codon control. The parameter combination that best recapitulated the control behavior in Fig 2C is displayed in Table 1. Error bars for simulated data are smaller than data markers.

slightly reduces uORF2 repressiveness by siphoning scanning ribosomes away from uORF2, and uORF3 is irrelevant for repression [31].

We inserted the uORF2-containing *UL4* leader sequence into a dual-luciferase reporter system (Fig 2B) in which nanoluciferase (NLuc) signal provides a readout of uORF2 repressiveness and firefly luciferase (FLuc) signal serves as normalization for transfection efficiency. This experimental platform can detect differences in luciferase activity over a 1000-fold range (S1 Fig). We confirmed that uORF2 repressiveness depends on its translation and the terminal diproline-dependent elongating ribosome stall (Fig 2C). Near-cognate start codons within uORF2 do not contribute to the uORF2 repressiveness (S1 Fig). We used this *UL4*-based luciferase reporter to quantitatively dissect the kinetics of uORF2-mediated translational regulation.

We complemented our experimental measurements with computational kinetic modeling of proposed models of uORF regulation (Fig 1). We aimed to find unique modeling predictions that would allow us to experimentally distinguish between the different models of uORF regulation. We specified the kinetics of each of the proposed models of uORF regulation using PySB, a framework for compact specification of rule-based models [58]. We then expanded the model into the BioNetGen modeling language syntax [59] and inferred a reaction

dependency graph for efficient simulation [39]. Next, we stochastically simulated the models using an agent-based Gillespie algorithm implemented in NFSim [60]. The molecules and reactions within the kinetic model are shown in Fig 3A and 3B, respectively, and are described in detail in the Materials and Methods section. We experimentally tested predictions from this computational modeling and used the results to refine our model specifications. This iterative cycle of experimental testing and computational modeling constituted our platform for differentiating between proposed uORF regulatory models.

To derive estimates for unknown parameters ('This work' in Table 1), we first calibrated our computational models to our reporter measurements on wild-type or mutant uORF2 (Fig 2C). We did not fit the constitutive repression and regulated re-initiation models (Fig 1A and 1E) to our reporter measurements (Fig 2C) since these models cannot account for the critical role of the *UL4* uORF2 elongating ribosome stall in regulating main ORF translation in single uORF-containing mRNAs. We used previously generated estimates for kinetic parameters not directly inferred in our work (Table 1).

Simulations of the queuing-mediated enhanced repression (Fig 1C) and collision-mediated 40S dissociation (Fig 1D) models readily recapitulate measurements of NLuc protein output from wild-type and mutant *UL4* reporters (Fig 2D, triangles and squares). The 80S-hit dissociation model (Fig 1B), modified to include an elongating ribosome stall within the uORF, also recapitulates the reporter measurements (Fig 2D, circles). However, this modified 80S-hit dissociation model requires the difference between the stronger Kozak and wild-type uORF initiation fractions to be quite large (80% vs. 2% compared to 50% vs. 10% for 2 other models mentioned above, Table 1). The derived ribosome loading rates (~0.02/s for all three of these models (Fig 1B–1D) are in line with literature estimates [52–54]. The re-initiation fractions derived here (50–70%, Table 1) are within the range of measured values across mRNAs with different sequence features [40–43]. A complete description of the derivation of model parameters can be found in the Materials and Methods section.

## Computational modeling predicts that different models of uORF regulation have unique parameters important for buffering

Following calibration of our computational models to recapitulate experimental data, we used these models to predict how translation would be perturbed upon varying other kinetic parameters. While many kinetic parameters could be varied to help distinguish between proposed models of uORF regulation (Fig 1), we honed in on the rate of ribosome loading onto the mRNA for two key reasons. Firstly, this rate is reduced endogenously in response to a variety of cellular and environmental signals. Amino acid deprivation, ribosome collisions, dsRNA viral infection, unfolded proteins, and heme deprivation are sensed by one of the four eIF2α kinases (GCN2, PKR, PERK, and HRI) to reduce TC concentration [61–64]. A reduction in the concentration of eIF2α-containing TCs reduces the rate of ribosome loading. Viral infection also leads to reduced ribosome loading via interferon-induced proteins with tetratricopeptide repeats (IFITs) [65]. Cellular stress also reduces ribosome loading via inhibition of mTOR and sequestration of eIF4E by hypophosphorylated 4EBP [66]. Secondly, translated repressive uORFs are enriched in transcripts buffered against reduced ribosome loading [67–70]. Therefore, we were particularly interested in varying this ribosome loading rate to investigate if and how uORFs provide this buffering across various proposed models. For each of the five surveyed models of uORF regulation (Fig 1), we investigated what uORF parameter combinations, if any, allow buffering against reduced ribosome loading rates.

We use the term 'buffer' to describe the observation of main ORF protein output decreasing less than expected, or even increasing, with reduced ribosome loading in comparison to the

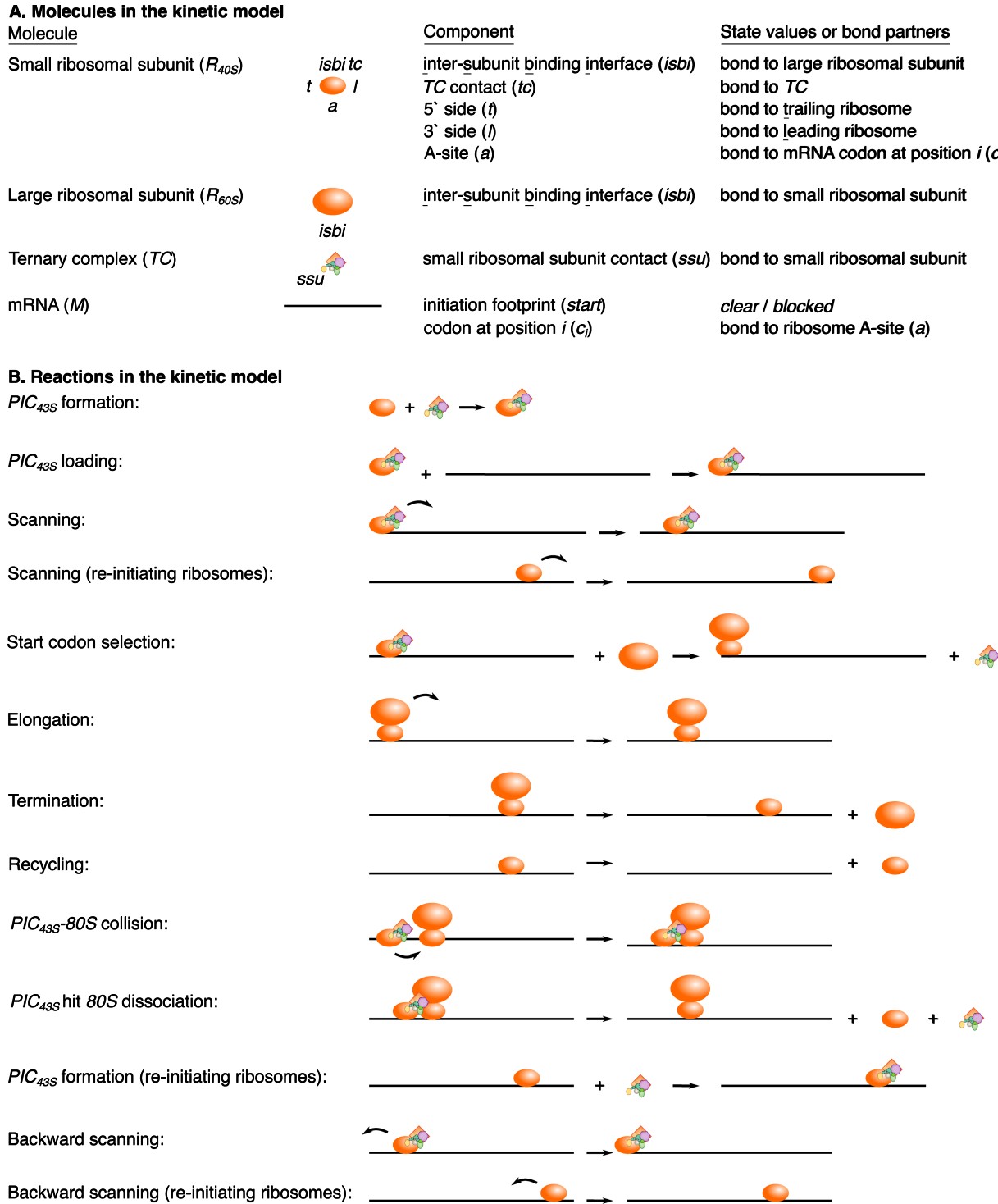

**Fig 3. Modeling workflow. (A)** Molecules in the kinetic model. Molecules have components each of which has state values or binding sites for other molecules (called *bond* in BioNetGen). Components are abbreviated in parentheses by how they are referenced in the model specification. For example, the mRNA (*M*) initiation footprint ($c_1$ to $c_n$ where n is equal to the ribosome footprint size in nt) can either be *clear* of ribosomes, and therefore free for a $PIC_{43S}$ loading reaction to occur, or *blocked* by a ribosome, preventing this reaction. **(B)** Visual representations of the reactions in the kinetic model. Re-initiation necessitates several additional reactions. $PIC_{43S}$ formation ($R_{40S}$ binding *TC*) can occur if the $R_{40S}$ is bound to the mRNA; this *TC* re-binding is required for start codon selection competence. $R_{40S}$ molecules can scan forward or backward. Some reactions in the kinetic model, such as different types of collision and dissociation reactions, are not depicted here.

**Table 1. Parameter ranges and fit values for modeling.**

| Parameter | Value range | Fit value (80S-hit dissociation) | Fit value (queuing- mediated enhanced repression) | Fit value (collision- mediated 40S dissociation) | Reference |
|---|---|---|---|---|---|
| $k_{cap\ bind}$ (s$^{-1}$) | 0.02–0.06 | 0.016 | 0.023 | 0.025 | This work [52–54] |
| $k_{scan}$ (nt/s) | 1–10 | 5 | 5 | 5 | [38,55] |
| $k_{start\ uORF2\ WT}$ (s$^{-1}$) | unknown | 0.1 | 0.5 | 0.5 | This work |
| WT uORF2 initiation (%) | unknown | 2 | 10 | 10 | This work |
| $k_{start\ uORF2\ strong\ Kozak}$ (s$^{-1}$) | unknown | 20 | 5 | 5 | This work |
| strong Kozak uORF2 initiation (%) | unknown | 80 | 50 | 50 | This work |
| $k_{elong}$ (codons/s) | 3–10 | 5 | 5 | 5 | [52–54,56] |
| $k_{elong\ stall}$ (codons/s) | 0.001 | 0.001 | 0.001 | 0.001 | [57] |
| $k_{terminate}$ (s$^{-1}$) | 0.5–5 | 1 | 1 | 1 | [56] |
| $k_{terminated\ ssu\ recycle\ uORF}$ (s$^{-1}$) | unknown | 2 | 5 | 5 | This work |
| Re-initiation (%) | unknown | 75 | 50 | 50 | This work |
| $k_{dissociate}$ (s$^{-1}$) | unknown | 2 | 0 | 2 | This work |
| uORF length (codons) | 21 | 21 | 21 | 21 | [32] |

constitutive repression model (Fig 1A). The constitutive repression model (Fig 1A) has no buffering (Fig 4A) since its repression is independent of the ribosome loading rate. Buffering requires an interaction between ribosome loading and the degree of translational repression. We use buffering as an overarching term that encompasses both resistance and preferred translation. Resistance refers to a decrease in main ORF protein output to a lower extent than in the constitutive repression model when ribosome loading is reduced. Preferred translation refers to increased main ORF protein output when ribosome loading is reduced.

The 80S-hit dissociation model (Fig 1B) displays buffering (Fig 4B, left panel, yellow-green line) in agreement with previous work [38]. This behavior arises because the number of 5′ elongating ribosomes that collide with scanning ribosomes correlates with the ribosome loading rate. However, buffering requires strong uORF initiation, minimal re-initiation, and a long uORF (Fig 4B, left panel, yellow-green line, S2A Fig) as observed previously [38]. These observations can be rationalized as follows. Strong uORF initiation generates sufficient elongating ribosomes that hit and knock off 3′ scanning ribosomes. Minimal re-initiation prevents the many uORF-translating ribosomes from also translating the main ORF. Longer uORFs offer more time for elongating ribosomes to catch up, hit, and knock off 3′ scanning ribosomes. However, most eukaryotic uORFs only weakly initiate translation and are short [1,3,4,8–11,31]. *UL4* uORF2 is 22 codons long, and we estimate re-initiation to be frequent (Table 1). Accordingly, buffering is no longer observed (S2B Fig) in the 80S-hit dissociation model when parameters (Table 1) derived from control *UL4* variants (Fig 2C) are used.

The queuing-mediated enhanced repression model [23] (Fig 1C) displays buffering behavior (Fig 4C, left panel, purple line) since the number of scanning ribosomes that initiate translation at the uORF is dependent on the rate of ribosome loading. In this model, reduced ribosome loading decreases the average queue length of ribosomes behind the elongation stall and, thus, also the fraction of ribosomes that initiate at the uORF2 start codon (S2C Fig, left). Unlike the 80S-hit dissociation model (Fig 1B), weakly initiating uORFs, such as *UL4* uORF2, still confer buffering in the queuing-mediated enhanced repression model (Fig 4C, left panel, purple line).

In the queuing-mediated enhanced repression model, enhanced uORF initiation and, therefore, buffering are sensitive to the distance between the uORF start codon and the

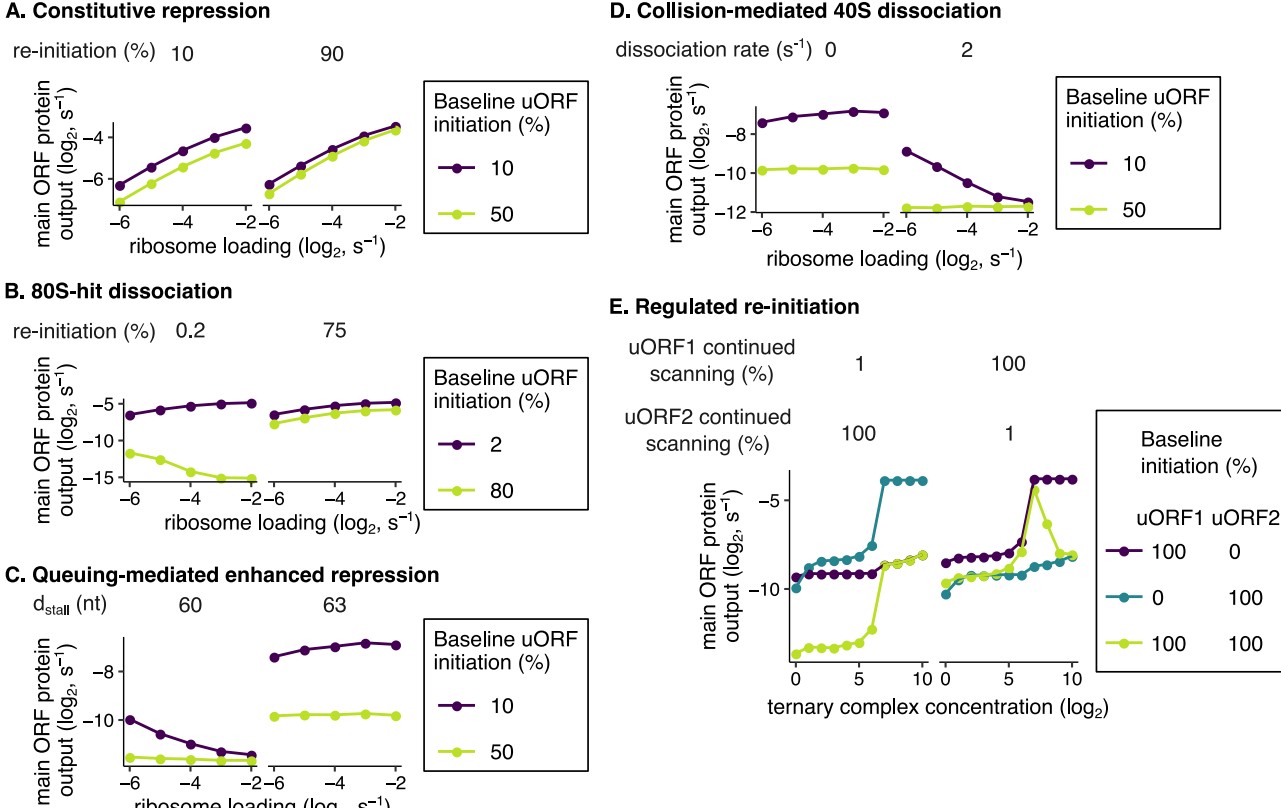

**Fig 4. Kinetic modeling predicts translational buffering by uORFs.** Buffering refers to a smaller than expected decrease (small positive slope), or even increase (negative slope), in main ORF protein output with reduced ribosome loading. **(A)** The constitutive repression model, without an elongating ribosome stall, has no buffering behavior. uORFs simply siphon away scanning ribosomes from the main ORF. **(B)** Buffering in the 80S-hit dissociation model depends on uORF initiation and re-initiation frequencies [38]. For buffering to occur in this model, uORFs must initiate well enough to have elongating ribosomes hit 3′ scanning ribosomes (yellow-green line). uORFs must also not continue scanning at high frequencies following termination (left panel); frequent continuation of scanning coupled with high uORF initiation allows many scanning ribosomes to make it to the main ORF. Buffering occurs better for longer uORFs that have more time for elongating ribosomes to hit 3′ scanning ribosomes (S2A Fig, yellow-green line). Here, the uORF is 100 codons long. The dissociation rate is 200s⁻¹, so 99% of scanning ribosomes hit by 5′ elongating ribosomes dissociate rather than continue scanning. The scanning rate is 2 nucleotides/s, and the elongation rate is 2 codons/s. There is no elongating ribosome stall in this model. **(C)** Buffering in the queuing-mediated enhanced repression model depends on $d_{stall}$: the distance between the uORF start codon and elongating ribosome stall. In this model, uORF initiation can increase above baseline with increased ribosome loading when $d_{stall}$ is an integer multiple of the ribosome footprint (30 nt, left panel). When this condition is met, buffering occurs. For $d_{stall}$ values of 60 and 63 nt, the uORF length is 21 and 22 codons, respectively. **(D)** Buffering in the collision-mediated 40S dissociation model depends on the dissociation rate. Here, $d_{stall}$ is 63 nt; with a low dissociation rate, this model reduces to the queuing-mediated enhanced repression model. **(E)** Buffering in the regulated re-initiation model depends on uORF initiation and continued scanning frequencies. For buffering to occur, several conditions must be met. At least 2 uORFs are required, both of which must be well-translated (yellow-green line). Continued scanning following termination at the first uORF must be frequent, and continued scanning following termination at the second downstream uORF must be rare (right panel). The second downstream uORF is 3 codons long. There is no elongating ribosome stall in this model. uORFs are located 25 nt from the 5′ cap. 99% of scanning ribosomes that make it to the main ORF will initiate translation; 1% will leaky scan. Unless otherwise stated, parameters (Table 1) obtained from calibrating models to reporter measurements on wild-type or mutant uORF2 (Fig 2C) are used here. Ribosome loading is the $k_{cap\,bind}$ rate for non-regulated re-initiation models. We model changes in ribosome loading via changes in $k_{cap\,bind}$ as that rate is easier to match to *in vivo* estimates of ribosome loading. However, buffering in the regulated re-initiation model is dependent on an eIF2α phosphorylation mechanism; we instead vary the number of ternary complexes in this model. Error bars of simulated data are smaller than data markers.

elongating ribosome stall ($d_{stall}$) (S2C Fig and Fig 4C, left vs. right panels). This sensitivity arises because $d_{stall}$ determines if the P-site of a queued scanning ribosome is correctly positioned at the uORF start codon to productively increase uORF initiation (S2C Fig, left). In the idealized case of homogeneously sized ribosomes (30 nt footprints [56,71]) and strict 5′-3′ scanning, $d_{stall}$ must be an integer multiple of 30 nt for buffering to occur. This strong

dependence of buffering on $d_{stall}$ is relaxed when backward scanning [41,72–74] occurs with a high rate (S2D Fig, middle). To simplify our modeling interpretations, we considered *UL4* uORF2, that is 22 codons long, to be 21 codons so that a queue behind the terminating ribosome stall positions a scanning ribosome's P-site exactly at the start codon.

The collision-mediated 40S dissociation model (Fig 1D) displays buffering (Fig 4D, right panel, purple line) because the number of scanning ribosomes that collide with 3′ stalled ribosomes depends on the rate of ribosome loading. Buffering in this model requires the collision-induced 40S dissociation rate to be somewhat fast (Fig 4D, right vs. left panels, S2E Fig, teal and yellow-green lines). If this rate is too low (for example, 0 in Fig 4D, left panel), this model reduces to the queuing-mediated enhanced repression model (Fig 1C). With an appreciable dissociation rate, the collision-mediated 40S dissociation model is not sensitive to the distance between the stall and the start codon (S2F Fig, purple lines). As in the queuing model (Fig 1C), weakly initiating uORFs, such as *UL4* uORF2, can still confer buffering (Fig 4D, right panel, purple line) in the collision-mediated 40S dissociation model (Fig 1D). This effect arises because, unlike in the 80S-hit dissociation model (Fig 1B), the elongation stall is now rate-limiting for main ORF translation. Therefore, an elongation stall in the collision-mediated 40S dissociation model or in the queuing-mediated enhanced repression model with permissive $d_{stall}$ spacing imparts buffering.

In the regulated re-initiation model (Fig 1E), buffering is observed (Fig 4E, right panel, yellow-green line) because termination at the first uORF stop codon is followed by re-initiation at either the second downstream uORF or the main ORF depending on the ternary complex concentration. Buffering in the regulated re-initiation model (Fig 1E) depends on the initiation efficiency and continued scanning fraction (fraction of terminating but non-recycling ribosomes) of the two uORFs (Fig 4E). Continued scanning following termination at the first uORF must be frequent while continued scanning following termination at the second downstream uORF must be rare. Higher ternary complex concentrations bias towards initiation at the second downstream uORF (S3A Fig). Reductions in ternary complex concentrations bias towards main ORF initiation; therefore main ORF translation can increase with decreased ribosome loading.

As such, our computational results provide the first systematic dissection of different mechanisms of uORF-mediated regulation (Fig 1) and enable their comparison with experimental measurements below.

### *UL4* uORF2 buffers against reductions in main ORF protein output from reduced ribosome loading in an elongating ribosome stall-dependent manner

We next tested whether the computational predictions of uORF-mediated buffering (Fig 4) can be observed experimentally with *UL4* uORF2. To this end, we experimentally varied the rate of ribosome loading and measured effects on main ORF protein output using our reporter system (Fig 2B). Since the no-stall uORF2 variants have similar protein expression to the no-start uORF2 variants (Fig 2C), luciferase signal from the no-stall uORF2 variants provides a readout of the ribosome loading rate. If buffering were absent, then we would expect NLuc translation to be reduced equally between the no-stall and wild-type variants when ribosome loading is reduced.

We used three strategies to vary the rate of ribosome loading. We first used stem-loops near the 5′ cap to reduce the rate of 43S-cap binding without affecting mRNA stability (Fig 5A) [75,76]. We varied the degree to which ribosome loading is reduced by altering the GC content of the stem-loops; generally, higher GC content stem-loops are more stable and therefore

cause greater reductions in ribosome loading. We observe that NLuc signal decreases less with reduced ribosome loading for the wild-type *UL4* reporter in comparison to the no-stall UL4 variant (Fig 5A, left panel, yellow vs. gray circles). Therefore, NLuc protein output is resistant to stem-loop-mediated reduction in ribosome loading. When the wild-type data are normalized by the no-stall data, NLuc translation negatively correlates with ribosome loading, indicative of buffering against reduced ribosome loading by wild-type uORF2 (Fig 5A, right panel).

We also reduced ribosome loading with the drug thapsigargin, which induces the integrated stress response (ISR) by triggering ER stress (Fig 5B) [79]. We added a PEST tag [78] to increase the turnover of the NLuc protein to more accurately measure changes in main ORF translation during drug treatment. NLuc protein output from the wild-type *UL4* reporter decreases less in comparison to the no-stall control upon thapsigargin treatment (Fig 5B, left panel, yellow vs. gray circles), indicative of resistance. Again, when the wild-type data are normalized by the no-stall data, we observe that NLuc translation negatively correlates with ribosome loading, indicative of buffering against reduced ribosome loading by wild-type uORF2 (Fig 5B, right panel).

Finally, we added a short, synthetic uORF, 5′ to the *UL4* uORF2, to siphon scanning ribosomes away from uORF2 (Fig 5C). We varied the degree of ribosome siphoning by varying the Kozak context of the synthetic uORF, which in turn determines the rate of ribosome loading onto the uORF2-NLuc portion of the mRNA. Here, we observe that more NLuc is produced from the wild-type *UL4* reporter as scanning ribosomes are increasingly siphoned off by improving the Kozak context of the synthetic uORF (Fig 5C, left panel, yellow circles). While resistance is observed with the other strategies of reduced ribosome loading (Fig 5A and 5B, left panels), preferred translation is observed here (Fig 5C, left panel), perhaps because ribosome loading is reduced at the scanning step instead of at the cap-binding step. Similar to the other two strategies, when the wild-type data are normalized by the no-stall data, NLuc translation negatively correlates with ribosome loading, indicative of buffering against reduced ribosome loading by wild-type uORF2 (Fig 5C, right panel).

## Distance between the start codon and the stall does not systematically regulate uORF repressiveness or buffering

Given our experimental data demonstrating uORF2-mediated buffering of *UL4* reporters (Fig 5), we narrowed our focus from the five surveyed models (Fig 1) to the two (Fig 1C and 1D) most relevant for *UL4* uORF2: the queuing-mediated enhanced repression (Fig 1C) and collision-mediated 40S dissociation (Fig 1D) models. These two models are computationally predicted to confer buffering in an elongating ribosome stall-dependent manner without needing multiple uORFs (Fig 4C and 4D). To differentiate between these models, we turned to our computational modeling prediction that, only in the queuing-mediated enhanced repression model (Fig 1C), main ORF protein output is sensitive to the distance between the uORF start codon and the elongating ribosome stall (Fig 4C). Our computational modeling of the queuing-mediated enhanced repression model (Fig 6A, yellow-green line) predicts two broadly spaced clusters of main ORF protein output. Protein output from the main ORF is repressed when the start codon-stall distance is an integer multiple of the ribosome size. Protein output from the main ORF is high when the start codon-stall distance is not an integer multiple of the ribosome size. In contrast, the collision-mediated 40S dissociation model (Fig 1D) predicts a much lower effect of $d_{stall}$ on uORF repressiveness (Fig 6A, left panel, purple line). The residual effect of $d_{stall}$ on uORF repressiveness (Fig 6A, left panel, purple line) in the collision-mediated 40S dissociation model (Fig 1D) arises because the dissociation rate is low enough to allow rare queuing.

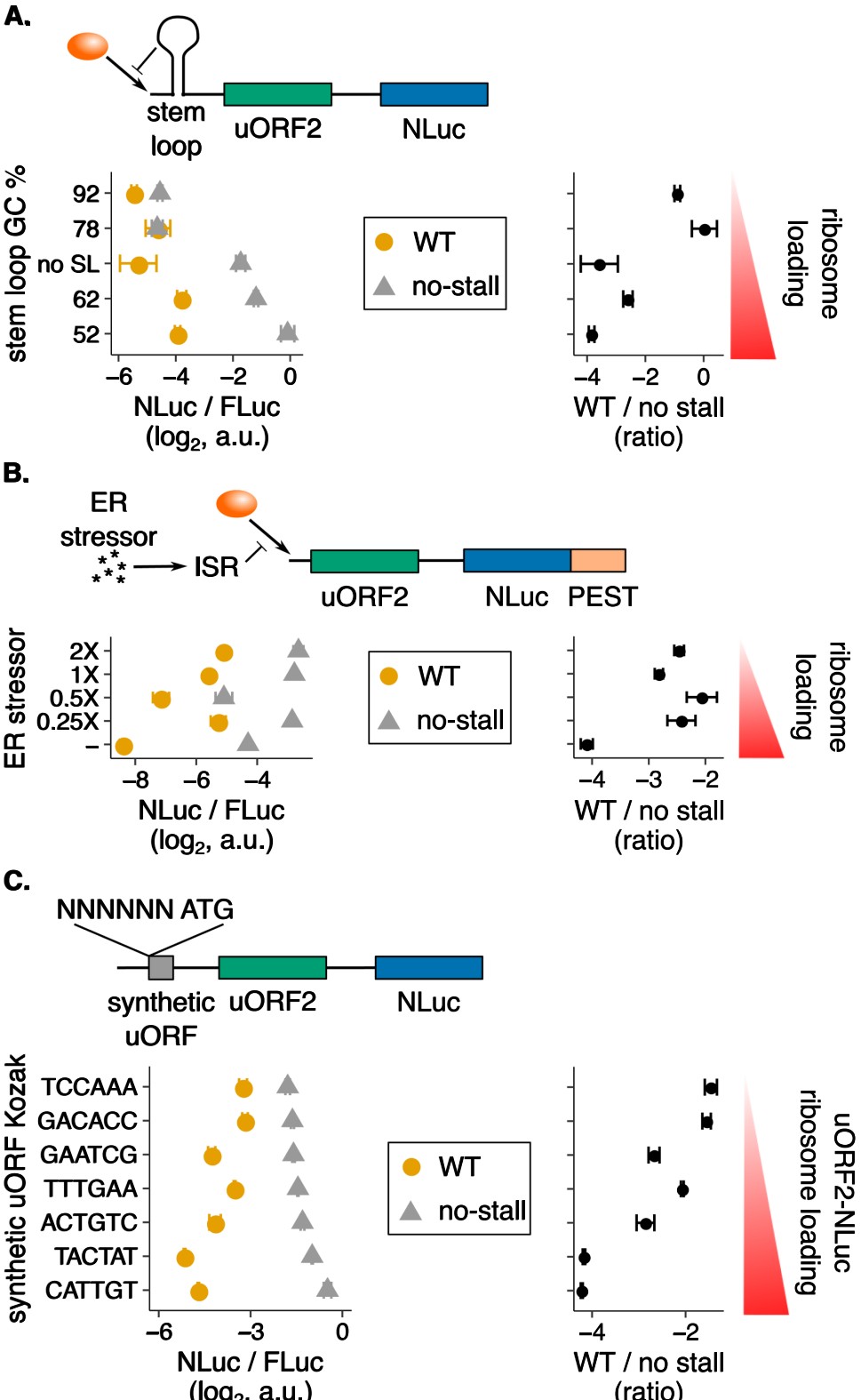

**Fig 5. The human cytomegaloviral uORF2 buffers against reductions in main ORF protein output.** The human cytomegaloviral *UL4* uORF2 is used in the dual-luciferase assay (Fig 2B) in conjunction with three experimental strategies to reduce ribosome loading. **(A)** Ribosome loading is reduced using stem-loops [76] with the indicated GC percentages. All stem-loops are positioned 8 nt from the 5′ cap and have the same predicted stability of -30 kcal/mol.

The no-stem-loop construct has a CAA repeat instead of a stem-loop. The 5′ UTR is 287 nt long. Data are normalized to a no-uORF start codon control without a stem-loop. **(B)** Ribosome loading is reduced using the drug thapsigargin (1X = 1 μM) [77], which induces the integrated stress response (ISR) by triggering ER stress. NLuc has a C-terminal PEST tag to turnover [78] of protein produced prior to the 6-hour drug treatment. The 5′ UTR is 236 nt long. Data are normalized to a no-uORF start codon control without a PEST tag. Error bars show standard error of mean NLuc / FLuc ratios over 4 biological replicates. **(C)** Ribosome loading onto the uORF2-NLuc portion of the transcript is reduced using a 5′ synthetic uORF: ATG GGG TAG. The synthetic uORF Kozak is varied to alter ribosome loading. The variants are vertically ordered by the no-stall means. The 5′ UTR is 262 nt long. Data are normalized to a no-uORF start codon control without a synthetic uORF. Right panels in A,B, C show wild-type (WT) mean values normalized by the corresponding no-stall values. The no-stall uORF2 mutants lack their terminal diproline motifs (P22A mutation). Unless stated otherwise, error bars show standard error of mean NLuc / FLuc ratios over 3 biological replicates.

Backward scanning is predicted to diminish the periodicity in main ORF translation with varying $d_{stall}$ lengths in both models (Fig 6A). However, backward scanning occurring as fast as forward scanning (~ 5 nucleotides/s) is required to abolish the periodicity in the queuing model (Fig 6A, right panel, yellow-green line). While there are estimates of how far ribosomes can backward scan [72–74], we are not aware of any backward scanning rate estimates. It is unlikely that the rate of backward scanning approaches the rate of forward scanning (5 nucleotides/s here) given the 5′-3′ directionality of scanning. Slower backward scanning (~ 3.75 nucleotides/s) is sufficient to abolish periodicity in the collision-mediated 40S dissociation model (Fig 6A, middle panel, purple line). This effect is not surprising given that the presence of periodicity in the latter model arises from rare queuing behavior. Therefore, our computational predictions of greater periodicity in main ORF translation across varied $d_{stall}$ in the queuing model hold even with backward scanning.

We then experimentally varied the distance between the start codon and the stall of the *UL4* uORF by adding codons to the 5′ end of uORF2. With *EYFP* donor sequences, we observe less than 2-fold changes in translational regulation (Fig 6B, top 7 rows) with no systematic trend with variations in uORF2 length, which is inconsistent with computational modeling predictions of the queuing-mediated enhanced repression model (Fig 6A, left panel). We observe similar results with a different donor sequence (S4 Fig, top 7 rows), confirming the generality of the observed repression with changes in uORF2 length. With both donor sequences, the longest uORF mutants are less repressive, but this effect may be due to decreased elongating ribosome stall strength. In these cases, the longer nascent peptides can extend out of the exit tunnel and can be bound by additional factors [27,28] or cotranslationally fold to exert a pulling force [80] to relieve the stall. Thus, in summary, varying the length of the UL4 uORF2 stall does not match computational predictions for sensitivity of main ORF repression to $d_{stall}$ in the queuing-mediated enhanced repression model (Fig 1C) and better supports the collision-mediated 40S dissociation model (Fig 1D).

In the queuing-mediated enhanced repression model (Fig 1C), buffering is uniquely predicted to be sensitive to the distance between the uORF start codon and the elongating ribosome stall (Fig 4C). We, therefore, asked whether or not buffering would still be experimentally observed with a disruption in this distance. Using our synthetic uORF method of reducing ribosome loading (Fig 6C), we observe that a 6 nt longer $d_{stall}$ uORF still buffers against reduced ribosome loading (Fig 6C, top two rows compared to bottom two rows). Since backward scanning of 15–17 nt has been observed [72–74], one would expect that buffering would still be predicted in the queuing model even with an increase in $d_{stall}$ of 6 nt. However, our computational modeling predicts that even very fast backward scanning does not restore buffering when $d_{stall}$ is disrupted by 6 nt (S2D Fig, right). Thus, our experimental data does not match computational predictions of buffering sensitivity to $d_{stall}$ in the queuing-mediated

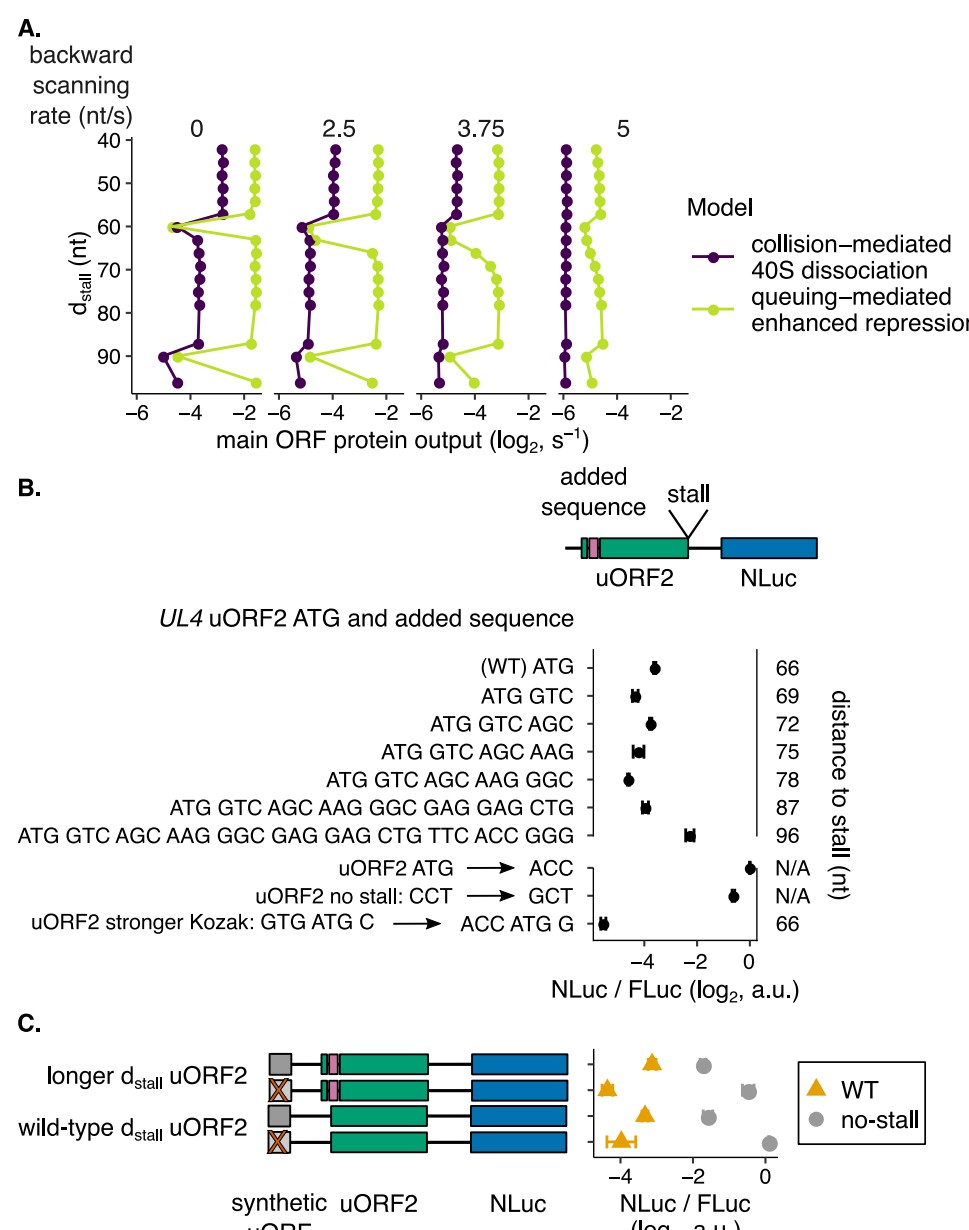

**Fig 6. Changes to the distance between the human cytomegaloviral uORF2 start codon and the elongating ribosome stall do not change repressiveness or buffering, consistent with the collision-mediated 40S dissociation model.** (A) Computational modeling predicts greater changes in uORF repressiveness with changes in $d_{stall}$ in the queuing-mediated enhanced repression model. Fast backward scanning abolishes this periodicity. $d_{stall}$ refers to the distance between the start codon and the elongating ribosome stall. As backward scanning increases in rate (moving right along panels), the collision-mediated enhanced repression model loses periodicity (middle panel, purple line) before the queuing-mediated enhanced repression model (right panel, yellow-green line). Parameters that best recapitulated reporter measurements on wild-type or mutant uORF2 (Fig 2C and Table 1) are used here. The forward scanning rate is 5 nucleotides/s. Data are normalized to a no-uORF start codon control. Error bars of simulated data are smaller than data markers. (B) Experimentally varying the distance between the human cytomegaloviral uORF2 start codon and the elongating ribosome stall does not systematically affect its repression of main ORF protein output. The human cytomegaloviral *UL4* uORF2 is used in the dual-luciferase assay (Fig 2B) in conjunction with various length inserts from the N-terminus of the *EYFP* main ORF. The *EYFP* main ORF sequence is inserted directly 3′ to the uORF2 start codon. The added sequence increases the distance between the uORF2 start codon and the elongating ribosome stall. The bottom three controls improve the uORF2 Kozak context, remove the start codon, and remove the elongating ribosome stall. Error bars show standard error of mean NLuc / FLuc ratios over 3 biological replicates. Data are normalized to the no-uORF start codon control. (C) Experimentally varying the human cytomegaloviral uORF2

$d_{stall}$ does not strongly regulate the capacity of buffering against reductions in main ORF protein output. Ribosome loading is reduced with a 5′ synthetic uORF: ATG GGG TAG. The no-stall uORF2 mutants lack their terminal diproline motifs (P22A mutation). No-synthetic uORF mutants (ATG to AAG) are depicted by transparent, gray bars with red Xs and have a higher relative ribosome loading rate onto the uORF2-NLuc portion of the transcript. The distance between the uORF2 start codon and the elongating ribosome stall is varied as indicated by adding 6 nt, GTC AGC, from the N-terminus of the *EYFP* main ORF. Data are normalized to a no-uORF start codon control without a synthetic uORF.

enhanced repression model (Fig 1C) but is consistent with the collision-mediated 40S dissociation model (Fig 1D).

## Several human uORFs have repressive terminal diproline motifs

Given that the elongating ribosome stall in the human cytomegaloviral *UL4* uORF2 is dependent on a terminal diproline motif, we asked whether there are other human uORFs similarly ending in diproline motifs that are also repressive. We searched for such uORFs in three databases: a comprehensive database of ORFs in induced pluripotent stem cells and human foreskin fibroblasts with 1,517 uORFs [6], a database integrated from *de novo* transcriptome assembly and ribosome profiling with 3,577 uORFs [81], and a database of proteins less than 100 residues in size derived from literature mining, ribosome profiling, and mass spectrometry with 1,080 uORFs [82]. We identified several human transcripts with terminal diproline-containing uORFs: *C1orf43*, *C15orf59*, *TOR1AIP1*, *PPP1R37*, and *ABCB9*. We replaced *UL4* uORF2 in our reporter (Fig 2B) with these human uORFs. We mutated the terminal proline codon to alanine codon as well as the start codon of these human uORFs and measured the effects of these mutations on NLuc protein output relative to the wild-type uORFs. While many of the tested uORFs are repressive (Fig 7, yellow vs. blue), unlike the human cytomegaloviral uORF2, these human uORFs still repress NLuc protein output without their terminal diproline motif (Fig 7, gray vs. blue), indicating additional contributions to translational repression from other residues in the nascent peptide and due to siphoning of scanning ribosomes at the start codon.

## Discussion

In this study, we use a combination of computational modeling and experimental reporter measurements to dissect the kinetics of uORF-mediated translational regulation of the *UL4* mRNA of human cytomegalovirus. We find that the elongating ribosome stall in *UL4* uORF2 buffers against reductions in main ORF protein output due to reduced ribosome loading (Fig 4). Using an experimentally-integrated modeling approach, we differentiate between models of regulation that can explain this observation. Our computational framework allows easy specification and efficient simulation of several previously proposed kinetic models of uORF regulation (Fig 1). While uORFs are enriched in stress-resistant transcripts, not all uORFs provide buffering [67]. We can predict which models of uORF regulation allow buffering and which parameters are key for buffering in each model (Fig 4). To our knowledge, our work is the first systematic investigation of what uORF metrics impart buffering in each kinetic model of uORF regulation.

uORFs are generally thought to simply siphon away scanning ribosomes from main ORFs, but this simple behavior in the constitutive repression model (Fig 1A) is not predicted to provide buffering (Fig 4C) [67–70]. Instead, we find that 5′ UTRs containing one (or some combination) of the following enable buffering of main ORF translation: scanning ribosome dissociation due to 80S hits from the 5′ end (Fig 1B), a single uORF with an elongating

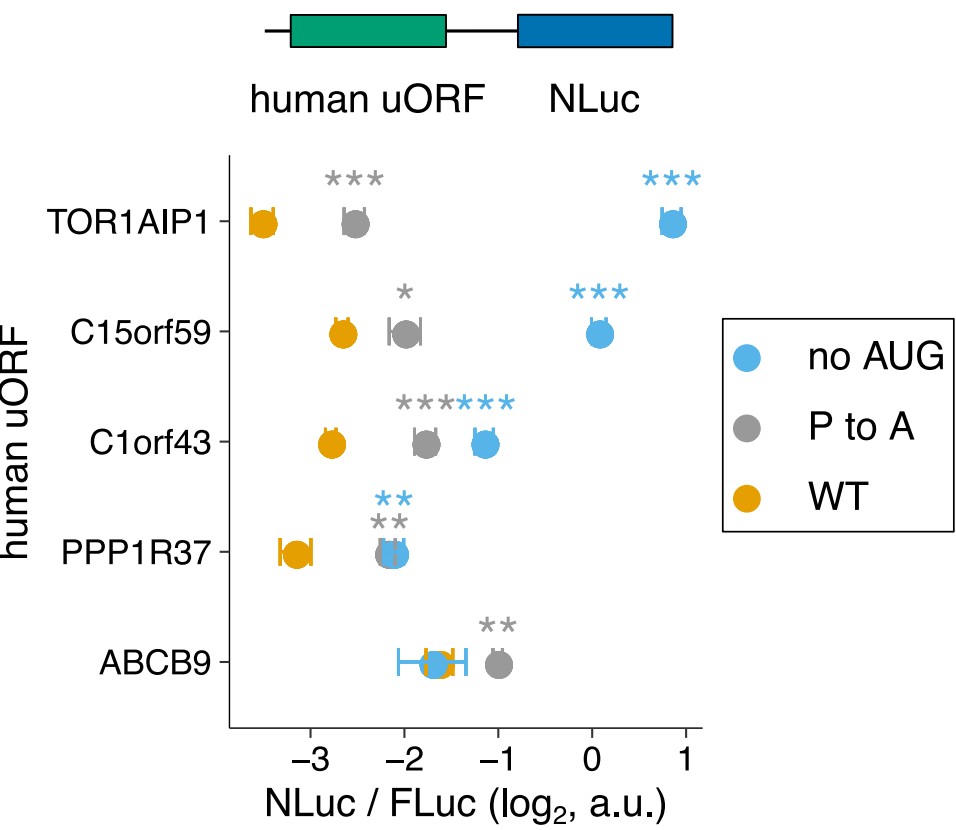

**Fig 7. Several human uORFs have repressive terminal diproline motifs.** Terminal diproline motif-containing human uORFs are used in the dual-luciferase assay (Fig 2B). The terminal proline codon in each uORF is mutated to an alanine codon in the P to A mutant. Start codons are mutated to ACC for the no-AUG mutants. P values comparing the indicated mutants to the wild-type are from a two sample t-test: * (0.01 < P < 0.05), ** (0.001 < P < 0.01), *** (P < 0.001). Error bars show standard error of mean NLuc / FLuc ratios over 5 biological replicates. Data are normalized to a no-*UL4*-uORF2 start codon control.

ribosome stall (Fig 1C and 1D), or multiple uORFs acting through the regulated re-initiation model (Fig 1E).

Long, well-initiating uORFs that do not re-initiate well allow buffering (Fig 4B, left panel, yellow-green line, S2 Fig panel A, yellow-green line) in the 80S-hit model (Fig 1B), but these requirements are at odds with the typically short and poorly initiating nature of known uORFs [1,3,4,8–11]. Indeed, when we use parameters specific to *UL4* uORF2 for the 80S-hit model (Table 1), namely that uORF2 initiates poorly, re-initiates well, and is not very long, buffering is no longer predicted (S2B Fig).

Computational predictions from the regulated re-initiation model (Fig 1E) agree (Figs 4E and S3A) with previous work [35,36] showing that buffering requires: 1) two well-translated uORFs and 2) frequent and rare continued scanning after termination at uORFs 1 and 2, respectively. Since 30% of human transcripts contain multiple uORFs, some of these might enable buffering by the regulated re-initiation model. However, about 25% of human transcripts only have one uORF [2] and cannot provide buffering under this model.

We narrowed our focus to the two models (Fig 1C and 1D) that are most pertinent to *UL4* uORF2. Both the queuing-mediated enhanced repression (Fig 1C) and collision-mediated 40S dissociation (Fig 1D) models are predicted to allow buffering (Fig 4C and 4D) with weakly initiating uORFs and elongating ribosome stalls. Both of these models require only a single uORF

for buffering (Fig 4C and 4D). Computational modeling not only predicts this buffering behavior but also allows us to differentiate between these two models. We predict that the queuing-mediated enhanced repression model (Fig 1C) is uniquely sensitive to the distance between the uORF start codon and elongating ribosome stall (Fig 6A, yellow-green line, Fig 4C, purple lines, S2F Fig, purple lines). We experimentally vary this distance and do not find any systematic changes in either main ORF protein output (Fig 6B) or buffering (Fig 6C). Based on our results, we propose that scanning ribosomes dissociate rather than queue when encountering a 3′ stalled elongating ribosome on uORF2 of *UL4* mRNA.

Scanning ribosomes have been predicted to dissociate upon encountering stable secondary structures on the mRNA [83]. Collisions between scanning ribosomes and their subsequent dissociation have also been proposed in a model of initiation quality control [84]. This dissociation could serve to maintain the free pool of 40S ribosomal subunits while still allowing regulation of main ORF translation. Collisions between scanning and elongating ribosomes and subsequent quality control are not well understood; what we describe as scanning ribosome dissociation here may be rescue by a quality control pathway.

Although our data from *UL4* uORF2 does not support the queuing-mediated enhanced repression model (Fig 1C) [23], this model might describe translation kinetics on other mRNAs. Translation from near-cognate start codons is resistant to cycloheximide, perhaps due to queuing-mediated enhanced initiation, but sensitive to reductions in ribosome loading [85]. Loss of eIF5A, which helps paused ribosomes continue elongation, increases 5′ UTR translation on human mRNAs with pause sites proximal to the start codon, perhaps also through queuing-mediated enhanced initiation [86]. There is also evidence of queuing-enhanced uORF initiation in the 23 nt long *Neurospora crassa* arginine attenuator peptide [87] as well as in transcripts with secondary structure near and 3′ to start codons [88]. Additional sequence elements in the mRNA might determine whether scanning ribosome collisions result in queuing or dissociation. Small subunit profiling data [89] from human uORFs that have conserved amino acid-dependent elongating ribosome stalls do not show evidence of scanning ribosome queues (S5A Fig), consistent with the collision-mediated 40S-dissociation model. However, subtle queues might not be observed given low read counts arising from insufficient capture of small ribosomal subunits in these experiments.

In our modeling, we assume homogenous footprint lengths of 30 nt for both scanning and elongating ribosomes. Even though heterogeneously sized footprints have been observed for small ribosomal subunits [89–91] and elongating ribosomes [92,93], our modeling of homogenous footprint length is appropriate for the following reasons. Firstly, with respect to the small ribosomal subunit footprints, crosslinking of associated eIFs is thought to be the main source of length heterogeneity [89,90], and homogenous 30 nt footprints are observed in the absence of crosslinking [90]. Secondly, in the context of the strong, minutes-long *UL4* uORF2 elongating ribosome stall [57], collided ribosomes, if they do not dissociate, will wait for long periods of time in a queue relative to normal scanning or elongating ribosomes, during which associated eIFs likely dissociate [90]. Thirdly, a sizable fraction of mRNAs exhibit cap-tethered translation in which eIFs must dissociate from ribosomes before new cap-binding events, and therefore collisions, can occur [90]. Elongating ribosome footprint heterogeneity is much less drastic than that observed for scanning ribosomes and likely arises from different conformational states such as empty or occupied A sites [92,93]. While different elongating ribosome footprints arise from differences in mRNA accessibility to nucleases, it is unclear whether the distance between two collided ribosomes changes across different ribosome conformations.

In addition to the *UL4* viral uORF studied here, several human uORFs are known to contain an elongating ribosome stall [22,23,26,47]. Apart from terminal diprolines, other motifs such as Arg-X-Lys at E-P-A sites [94] or specific dipeptides such as Gly-Ile, Asp-Ile, Gly-Asp

[95] can also cause elongation stalls. There are a variety of other mechanisms that may reduce the rate of elongation, such as mRNA stem-loops and G-quadruplexes [96], low tRNA availability [97,98], or interactions between the nascent peptide and the ribosome [99,100]. uORFs are often short [3] and may therefore be better poised to stall ribosomes since the nascent peptides might not be accessible to co-translational factors that pull the nascent peptide out of the ribosome [101,102]. Thus, a key role for elongating ribosome stalls in uORFs might be to enable buffering. While few uORF stalls have been mechanistically characterized [1], other elongating ribosome stall-containing uORFs, such as the ones in *MTR* [47] and *AMD1* [103] mRNAs, might enable buffering; the elongating ribosome stall-containing uORFs in *AZIN1* [23], *PPP1R15A* (*GADD34*) [104], and *DDIT3* (*CHOP*) [22] have already been shown to enable buffering. Conversely, uORFs in several single uORF transcripts known to buffer against stress, such as *SLC35A4*, *C19orf48*, and *IFRD1* [67], might act through elongating ribosome stalls.

The computational models considered here can be readily extended to incorporate more complex mechanisms of translational control. For example, in our models, initiation proceeds via a cap-severed mechanism in which multiple scanning ribosomes can be present in the 5′ UTR at the same time. If we were to model cap-tethered initiation, strong uORF elongating ribosome stalls would eventually sever this connection, similar to how the cap-eIF-ribosome connection is severed during the usually longer translation of main ORFs [90,105,106]. It will also be interesting to consider the effect of cellular stress-reduced elongation rates [107] and increased re-initiation [108], both of which might regulate uORF-mediated buffering, as well as elongating ribosome dissociation through known quality control pathways [39,84,109–114]. Translation heterogeneity among isogenic mRNAs has been observed in several single-molecule translation studies [33,52–54,115]. This heterogeneity may arise from variability in intrasite RNA modifications [116], RNA binding protein occupancy, or RNA localization. We do not capture these sources of heterogeneity in our modeling since the observables in our simulations are averaged over long simulated time scales and used to predict only bulk experimental measurements. However, the models studied here can readily be extended through compartmentalized and state-dependent reaction rates [59] to account for the different sources of heterogeneity observed in single-molecule studies.

## Materials and methods

### Plasmid construction

The parent cloning vector was created as follows. A commercial vector (Promega pGL3) with ampicillin resistance was used to clone NLuc and FLuc. NLuc expression is driven by a CMV promoter. FLuc expression is driven in the opposite direction within the plasmid and serves as an internal transfection control. The human cytomegaloviral *UL4* 5′ UTR was PCR amplified from HCMV genomic DNA. To create mutant 5′ UTR versions of the parent pGL3-FLuc-NLuc vector, the vector was digested with KpnI/EcoRI unless otherwise noted. 1 or 2 PCR-amplified fragments with 20–30 bp homology arms were then cloned using isothermal assembly [117]. The stem-loop [76] 5′ UTR mutants were cloned as follows. The stem-loops were ordered as oligonucleotides with overhangs for ligation into ClaI and NotI sites. The oligonucleotides were annealed and used in PCR reactions to add CMV homology arms. An AAVS1 parent vector was digested with ClaI and NotI. These stem-loops were then inserted into the ClaI/NotI restriction digested parent vector by isothermal assembly [117]. The stem-loops were then PCR amplified off of this plasmid and inserted into the pGL3-Fluc-*UL4*-5′-UTR-N-Luc parent vector described above. The several tested human uORFs were PCR amplified from human genomic DNA and inserted into a PstI/EcoRI digested parent. The inserted sequences were confirmed by Sanger sequencing. Kozak context and stall codon mutations were

introduced in the PCR primers used for amplifying inserts before isothermal assembly. Standard molecular biology procedures were used for all other plasmid cloning steps [118]. S1 Table lists the plasmids described in this study. Key plasmid maps are available at https://github.com/rasilab/bottorff_2022 as SnapGene.dna files. Plasmids will be sent upon request.

## Cell culture

HEK293T cells were cultured in Dulbecco′s modified Eagle medium (DMEM 1X, with 4.5 g/L D-glucose, + L-glutamine,—sodium pyruvate, Gibco 11965–092) and passaged using 0.25% trypsin in EDTA (Gibco 25200–056).

## Dual-luciferase reporter assay

Plasmid constructs were PEI or Lipofectamine 3000 (Invitrogen, L3000-008) transiently transfected into HEK293T cells for 12-16h in 96 well plates. After the 12-16h transfection, the ~110 μL media was removed and replaced with 20 μL media per well. The Promega dual-luciferase kit was used. Cells were lysed with 20 μL ONE-Glo EX Luciferase Assay Reagent per well for three minutes to measure firefly (*Photinus pyralis*) luciferase activity. Then, 20 μL NanoDLR Stop & Glo Reagent was added per well for 10 minutes to quench the FLuc signal and provide the furimazine substrate needed to measure NLuc luciferase activity. FLuc activity serves as an internal control for transfection efficiency, and NLuc activity provides a readout of 5′ UTR regulation of NLuc translation.

## Kinetic modeling

We specify our kinetic models using the PySB interface [58] to the BioNetGen modeling language [59] (Fig 3). The Python script is parsed by BioNetGen into a.bngl file and converted into an xml file for use as input to the agent-based stochastic simulator NFsim [60].

## Molecules

Our kinetic models of eukaryotic translational control describe the interactions between 3 molecule types: mRNA, ribosome (composed of separate large and small subunits), and ternary complex. Here, we describe these molecules' components, states, and binding sites (Fig 3A). mRNA molecules have the following components: 5′ end and codon sites ($c_i$). The mRNA 5′ end can either be free of (*clear*) or occupied with a ribosome (*blocked*). The mRNA 5′ end must be clear for a 43S to bind, which leaves the 5′ end blocked until the ribosome scans (or elongates) sufficiently 3′ downstream. The mRNA codon sites serve as binding sites for the ribosome A site. Small ribosomal subunits have the following components: inter-subunit binding interface (*isbi*), ternary complex contact (*tc*), 5′ side (*t* for trailing), 3′ side (*l* for leading), and A site (*a*). The inter-subunit binding interface site allows interactions between small and large ribosomal subunits; large ribosomal subunits also have the inter-subunit binding interface (*isbi*) components. The 5′ and 3′ side sites serve as binding sites for other ribosomes during collisions (5′ or 3′ side). The A site serves as a binding site for the mRNA. Both scanning and elongating ribosomes have mRNA footprints of 10 codons in our simulations based on mammalian ribosome profiling data [56,71]. Ternary complex molecules have a single component *ssusite* that serves as a binding site for the small ribosomal subunit.

## Reactions

We describe here each type of kinetic reaction in our models of eukaryotic translational control (Fig 3B). We use a syntax similar to that of BioNetGen [59] to illustrate the kinetic

reactions. We scale TC and ribosome subunit numbers (100 each) to the single mRNA present in the simulation. Simulation of a single mRNA over several rounds of translation is sufficient to infer steady state translation dynamics.

**Initiation: PIC (43S) formation.** Small ribosomal subunits must bind TCs to form pre-initiation complexes (PICs, 43Ss) before loading onto mRNAs. We assume that PIC formation is irreversible. PIC formation is not rate-limiting in our simulations; we set the rate of 43S-cap binding ($k_{cap\ bind}$) to be rate-limiting and to a total rate (independent of [43S]) to match the overall initiation rate to that of cellular estimates. Therefore, we arbitrarily set the second-order PIC formation rate (40S-TC binding rate, $k_{ssu\ tc\ bind}$) to $0.01\ ^*\ TC^{-1}\ ^*\ SSU^{-1}$ such that 100 40S-TC binding events occur per second, which is much higher than the 43S-cap binding rate.

**Initiation: PIC (43S) loading onto mRNA.** We model ribosome footprints at 30 nt following mammalian ribosome profiling data [56,71]. Therefore, PIC loading can occur when the 5′ most 30 nucleotides (nt) of the mRNA are not bound to any ribosome. The rate at which PICs load onto the 5′ end of the mRNA, $k_{cap\ bind}$, is varied over a 100-fold range from the maximum ribosome loading rate, 0.125/s, based on single-molecule estimations in human cells [52]. PICs can load onto the mRNA when a ribosome footprint-sized region at the 5′ mRNA end is free of ribosomes. PIC loading results in the 5′ end being blocked until this ribosome scans or elongates past a ribosome footprint from the 5′ cap. We assume that PIC loading is irreversible.

**Initiation: Scanning and start codon selection.** The scanning rate is 5 nucleotides/s following estimates in a mammalian cell-free translation system [55] and a previous computational study [38]. Small ribosomal subunit A sites must be positioned exactly over start codons to initiate translation. The uORF start codon is 25 nt from the 5′ cap. We vary the rate at which this start codon selection occurs at the uORF in our modeling. Start codon selection releases the TC bound to the small ribosomal subunit. We assume that TC is regenerated instantaneously. The start codon selection rate divided by the sum of this start codon selection rate, the scanning rate, and the backward scanning rate equals the baseline initiating fraction. This calculation of the baseline initiating fraction will underestimate the initiating fraction in the case of correctly positioned 3′ ribosome queues (as in the queuing-mediated enhanced repression model). We assume that start codon selection is irreversible.

**Elongation.** Elongation results in the ribosome A site moving from codon $c_i$ to codon $c_{i+1}$. The rate of elongation is set to 5 codons/s following single-molecule method and ribosome profiling estimates in mammalian cells of 3–18 codons/s [52–54,56,119]. Elongation may only proceed if there is no occluding 3′ ribosome; in other words, elongation may only proceed from codon $c_i$ to codon $c_{i+1}$ if the next 3′ ribosome's A site is bound to a codon no more 5′ than $c_{i+11}$. The elongation rate at the stall within the uORF is set to 0.001/s [57].

**Termination, continued scanning, and re-initiation.** Termination results in the dissociation of the large ribosomal subunit, but the small ribosomal subunit may continue scanning and subsequently re-initiate if a new TC is acquired before the next start codon is encountered. The termination rate is set to 1/s given that ribosome density tends to be higher at stop codons than within ORFs [56,92]. The recycling rate of terminated small ribosomal subunits after uORF translation is varied to model the effect of varied continued scanning after uORFs on the regulation of main ORF translation. The scanning rate divided by the sum of the scanning rate and this recycling rate equals the continued scanning fraction.

**Collisions and dissociations.** A collision between two ribosomes requires them to be separated by exactly one ribosome footprint in distance on the mRNA and results in binding between the 5′ side of the leading (3′ most) ribosome and the 3′ side of the trailing (5′ most) ribosome. Abortive (premature) termination of ribosomes results in their dissociation from

the mRNA and any collided ribosomes they are bound to. Different models have different non-zero dissociation rates. For instance in the 80S-hit model, the following rates are equal and non-zero: $k_{scan\ term\ 5\ hit\ 80s}$, $k_{scan\ term\ both\ hit\ 80s\ 80s}$, $k_{scan\ term\ both\ hit\ 80s\ 40s}$. These rates relate to the dissociation of scanning ribosomes upon collisions with a 5′ elongating ribosome. Both hit refers to collisions with ribosomes on both sides. In the collision-mediated 40S dissociation model, the following rates are equal and non-zero: $k_{scan\ term\ 3\ hit\ 40s}$, $k_{scan\ term\ 3\ hit\ 80s}$, $k_{scan\ term\ both\ hit\ 40s\ 40s}$, $k_{scan\ term\ both\ hit\ 40s\ 80s}$, $k_{scan\ term\ both\ hit\ 80s\ 40s}$, $k_{scan\ term\ both\ hit\ 80s\ 80s}$. These rates relate to the dissociation of scanning ribosomes upon collisions with a 3′ scanning or elongating ribosome. The in vivo abortive termination rates of scanning ribosomes are not known. Small ribosomal subunits that make it to the 3′ end of the mRNA through leaky scanning of all (u)ORFs always dissociate.

## Model calibration to reporter measurements

We derive the $k_{cap\ bind}$ rates by spline interpolation of computationally modeled protein output fit to experimental data (Fig 2C). We minimized the root mean square error between modeled protein output across variations in these parameters and the experimental data.

## Human uORF search

We import uORF lists from several databases [6,81,82]. The SmProt database [82] includes 3162 uORFs from ribosome profiling data, which we filter down first to 1080 uORFs after filtering for aligned matches, available Kozak context, near-cognate start codons, and non-duplicates. Two of these uORFs end in diproline motifs, including *C1orf43*. Another database is a set of high confidence ORFs derived from ribosome profiling of human-induced pluripotent stem cells (iPSCs) or foreskin fibroblast cells (HFFs) and was downloaded from https://www.ncbi.nlm.nih.gov/pmc/articles/PMC4720255/bin/NIHMS741295-supplement-3.csv [6]. This database includes 1517 high confidence (ORF-RATER score > 0.8) uORFs from either iPSCs or HFFs, which we filter down to 3 that end in diproline motifs, including *ABCB9*, *C1orf43*, and *TOR1AIP1*. The third database derives from HEK293T, HeLa, and K562 cells using ribosome profiling and was downloaded from https://static-content.springer.com/esm/art%3A10.1038%2Fs41589-019-0425-0/MediaObjects/41589_2019_425_MOESM3_ESM.xlsx [81]. This database includes 3577 uORFs which we filter down to 3 that end in diproline motifs and that are less than 60 codons in length for ease of cloning, including *ABCB9*,*C15orf59*, and *PPP1R37*.

## Supporting information

**S1 Fig.** **(A)** The indicated mutations not present in Fig 2C (the no NLUc start codon and no uORF2 near-cognate start codons) remove two adjacent NLuc ATG codons (ATGATG to ACCACC) and remove 4 uORF2 near-cognate start codons (CTG to CTA or TTG to TTA, red bars), respectively. The no NLuc start codon mutant abolishes NLuc signal, and the no uORF2 near-cognate start codons mutant does not greatly affect uORF2 repressiveness. Error bars show standard error of mean NLuc / FLuc ratios over 3 biological replicates. Data are normalized to the no-uORF start codon control.**(B)** Raw FLuc and NLuc signals for indicated mutations from Fig 2C. Mock refers to transfection of no plasmid. Multiple data points indicate biological replicates.
(EPS)

**S2 Fig.** **(A)** Buffering in the 80S-hit dissociation model is affected by uORF length. Re-initiation is 0.2%. uORF initiation is 80%. **(B)** Buffering in the 80S-hit dissociation model is lost

with control matched parameters. Buffering in the 80S-hit dissociation model requires strong uORF initiation and rare re-initiation (Fig 4B, left panel, yellow-green line) and is stronger with longer uORFs S2A Fig, yellow-green line). However, we estimate re-initiation to be frequent (Table 1) following calibration of our modeling (Fig 2D) to reporter measurements on wild-type or mutant uORF2 (Fig 2C). uORF initiation is 2%. When the elongating ribosome stall is present, $d_{stall}$ is 63 nt to prevent reduction to the queuing-mediated enhanced repression model. **(C)** Queuing-mediated enhanced uORF initiation is sensitive to $d_{stall}$. As the rate of ribosome loading increases, the average queue size increases and allows enhanced uORF initiation only when $d_{stall}$ equals an integer multiple of the ribosome footprint (30 nt). **(D)** Backward scanning only relaxes the dependence of buffering on $d_{stall}$ in the queuing-mediated enhanced repression model when $d_{stall}$ is close to an integer multiple of the ribosome footprint (30 nt). The forward scanning rate is 5 nucleotides/s. For $d_{stall}$ values of 60, 63, 66 nt, the uORF length is 21, 22, 23 codons, respectively. **(E)** Buffering in the collision-mediated 40S dissociation model occurs even with a rather low dissociation rate. Here, $d_{stall}$ is 63 nt. **(F)** Buffering in the collision-mediated enhanced repression model (Fig 1D) is insensitive to $d_{stall}$. All rates and labels are identical to Fig 4 unless otherwise specified. Error bars of simulated data are smaller than data markers.
(EPS)

**S3 Fig.** **(A)** Initiation at the second downstream uORF is dependent on high ternary complex concentration. Initiation at the first uORF is 100%. Continued scanning fractions at both uORFs are 100%. Following termination at the first uORF, initiation at the second downstream uORF depends on if a new ternary complex has been acquired since termination at the first uORF. Only when ternary complex concentration is high does this real uORF2 initiation fraction approach the predicted fraction. **(B)** With an elongating ribosome stall, the 80S-hit dissociation model acquires $d_{stall}$-dependent buffering similar to that in the queuing-mediated enhanced repression model (Fig 4C). Re-initiation is 0.2%. All rates and labels are identical to Fig 4 unless otherwise specified. Error bars of simulated data are smaller than data markers.
(EPS)

**S4 Fig.** Experimentally increasing the distance between the human cytomegaloviral uORF2 start codon and elongating ribosome stall using FLAG donor sequence. The human cytomegaloviral *UL4* uORF2 is used in the dual-luciferase assay (Fig 2B) in conjunction with various length inserts from the N-terminus of the *FLAG* main ORF. The *FLAG* main ORF sequence is inserted directly 3′ to the uORF2 start codon. The added sequence increases the distance between the uORF2 start codon and elongating ribosome stall. The bottom two controls improve the uORF2 Kozak context and remove the start codon. Error bars show standard error of mean NLuc / FLuc ratios over 3 biological replicates. Data are normalized to a no-uORF start codon control.
(EPS)

**S5 Fig.** Ribosome density within elongation stall-containing human uORFs. **(A)** Small ribosomal subunit (TCP-seq [89]) coverage data. **(B)** Elongating ribosome (Ribo-seq, A site global aggregate) coverage data. *AMD1*, *AZIN1*, *DDIT3* (*CHOP*), *MTR* and *PPP1R15A* (*GADD34*) uORF amino acid sequences are MAGDIS, IPPKKRRRFTRLFGPLSHGELSDQVYNYPEGL-GEVLYREQFDFNAEPPWEPS, MLKMSGWQRQSQNQSWNLRRECSRRKCIFIHHHT, MSRRPPLPVFSWVLFRAVPRLRLWPRVSGC, and MNALASLTVRTCDRFWQTE-PALLPPG, respectively. Elongation stall locations are marked with red arrows. Coverage data were downloaded from GWIPS [120].
(EPS)

**S1 Table. List of plasmids used in this study.**
(CSV)

# Acknowledgments

We thank members of the Subramaniam lab, the Basic Sciences Division, and the Computational Biology Program at Fred Hutch for assistance with the project and discussions and feedback on the manuscript. The computations described here were performed on the Fred Hutch Cancer Research Center computing cluster.

# Author Contributions

**Conceptualization:** Ty A. Bottorff, Adam P. Geballe, Arvind Rasi Subramaniam.

**Formal analysis:** Ty A. Bottorff, Arvind Rasi Subramaniam.

**Funding acquisition:** Arvind Rasi Subramaniam.

**Investigation:** Ty A. Bottorff, Arvind Rasi Subramaniam.

**Methodology:** Heungwon Park, Adam P. Geballe, Arvind Rasi Subramaniam.

**Project administration:** Adam P. Geballe, Arvind Rasi Subramaniam.

**Supervision:** Adam P. Geballe, Arvind Rasi Subramaniam.

**Writing – original draft:** Ty A. Bottorff, Arvind Rasi Subramaniam.

**Writing – review & editing:** Ty A. Bottorff, Adam P. Geballe, Arvind Rasi Subramaniam.

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
