## [Decision Letter · Decision Letter 0]

4 Oct 2022

Dear Dr Subramaniam,

We are pleased to inform you that your manuscript entitled "Translational buffering by ribosome stalling in upstream open reading frames" has been editorially accepted for publication in PLOS Genetics. Congratulations!

The manuscript was evaluated by two of the original referees from Review Commons. As you will see, both referees are enthusiastic about moving forward. Before your submission can be formally accepted and sent to production you will need to complete our formatting changes, which you will receive in a follow up email. Please be aware that it may take several days for you to receive this email; during this time no action is required by you. Please note: the accept date on your published article will reflect the date of this provisional acceptance, but your manuscript will not be scheduled for publication until the required changes have been made.

Yours sincerely,

Gregory Barsh

Editor-in-Chief

PLOS Genetics

Gregory Copenhaver

Editor-in-Chief

PLOS Genetics

Comments from the reviewers (if applicable):

Reviewer's Responses to Questions

**Comments to the Authors:**

Reviewer #1: The authors have addressed my comments in the reviewing round.

Reviewer #2: The authors have appropriately addressed my concerns.

**Have all data underlying the figures and results presented in the manuscript been provided?**

Reviewer #1: Yes

Reviewer #2: **No: **Data may have been provided, but I did not have access to spreadsheets underlying graphs or summary statistics.

PLOS authors have the option to publish the peer review history of their article (what does this mean?). If published, this will include your full peer review and any attached files.

Reviewer #1: No

Reviewer #2: No

**Data Deposition**

http://datadryad.org/submit?journalID=pgenetics&manu=PGENETICS-D-22-01034

**Press Queries**

---

## [Editor Report · Acceptance letter]

21 Oct 2022

PGENETICS-D-22-01034 

Translational buffering by ribosome stalling in upstream open reading frames 

Dear Dr Subramaniam, 

We are pleased to inform you that your manuscript entitled "Translational buffering by ribosome stalling in upstream open reading frames" has been formally accepted for publication in PLOS Genetics! Your manuscript is now with our production department and you will be notified of the publication date in due course.

With kind regards,

Anita Estes

PLOS Genetics

On behalf of:
